# Prediction of the Ultra-Low-Cycle Fatigue Damage of Q345qC Steel and its Weld Joint

**DOI:** 10.3390/ma12234014

**Published:** 2019-12-03

**Authors:** Qin Tian, Hanqing Zhuge, Xu Xie

**Affiliations:** 1School of Civil Engineering and Architecture, Nanchang University, Nanchang 330031, Jiangxi Province, China; tianqin224@163.com; 2School of Civil Engineering and Architecture, Zhejiang University, Hangzhou 310012, Zhejiang Province, China; 11512058@zju.edu.cn

**Keywords:** steel piers, weld specimen, ultra-low-cycle fatigue, continuous damage mechanics, prediction method for damage

## Abstract

Based on the continuum damage mechanics model (CDM) for monotonic tension, a new CDM for ultra-low-cycle fatigue (ULCF) is put forward to predict ULCF damage of steel and its weld joint under strong earthquakes. The base metal, heat-affected zone and weld metal of Q345qC steel were considered as research objects, and the uniaxial plastic strain threshold of the CDM model was calibrated via tensile testing combined with finite element analysis of notched round bar specimens. ULCF tests of the base metal and weld specimens were carried out to analyse their fatigue life, fracture life and post-fracture path. Based on the calibrated uniaxial plastic strain threshold, the finite element models of base metal and weld specimens suitable for CDM model were established by ABAQUS. The calibration results of material parameters show that the weld metal has the lowest plastic strain threshold and the largest dispersion coefficient at the plastic strain threshold. Prediction results under cyclic loading with a large strain were compared with experimental values, and results showed that the predicted crack initiation and fracture lives of the base metal and weld specimens are lower than their corresponding experimental values. The predicted errors of crack initiation life and fracture life decrease with increasing strain level. The development law of the damage variable reveals exponential growth combined with a stepped pattern. The CDM model can also accurately predict the number of cycles to initial damage. Taking the results together, the CDM of the ULCF of the base metal and weld specimens could successfully predict post-fracture paths.

## 1. Introduction

Steel piers are seismic-vulnerable components of bridge structures. In previous earthquakes, large plastic deformation and local buckling of steel plates led to seismic damage of steel piers [1] and ultra-low-cycle fatigue (ULCF) damage at welded joints [2]. For example, in the 1995 Kobe earthquake in Japan, ULCF damage was observed at the weld joints of steel structures [2,3]. Since this earthquake, researchers worldwide have widely studied the local buckling of steel plates and ULCF damage at welded joints. Some structural measures, such as setting longitudinal stiffened ribs, increasing the thickness of the steel plate and filling concrete into steel piers, have been used to solve the problem of local buckling failure [4,5,6]. However, the problem of ULCF damage has yet to be completely explored.

A number of researchers have recently carried out studies on the mechanism of the LCF and ULCF damage of steel materials [7,8]. The fracture mechanism of ULCF is essentially different from that of LCF. LCF fractures involve brittle fractures [9], whereas ULCF failures are due to ductile failure under the cyclic loading of a large strain [10,11]. ULCF failure is characterised by a few loading cycles (generally less than 20) [12,13,14]. For example, in 1998, Kuwamura [15] confirmed that the fracture morphologies of LCF and ULCF failures greatly differ by applying electron microscopy. ULCF cracks are blunt, wide and largely open, and the fracture surface shows obvious dimples; these characteristics indicate a ductile failure morphology [16]. By comparison, LCF cracks are sharp, narrow and deep and present a brittle fracture morphology. In addition, microvoids are not formed around the fractures LCF failure, and the fracture mode of this failure type is transgranular cleavage fracture [17]. Considering that the fracture mechanism of ULCF is different from that of LCF [9,18,19,20], damage prediction methods suitable for LCF cannot be directly used for ULCF damage prediction.

Several scholars have studied the prediction methods of fatigue life and damage process of ULCF in steel materials. At present, four main methods are used to study the fracture properties of steel and its weld joints, namely, the traditional fracture mechanics method, the empirical formula, the micro mechanical model and the continuum damage mechanics model (CDM). Early researchers mainly used fracture mechanics to study the fracture properties of steel. The traditional fracture mechanics method assumes that cracks exist and that the initial crack tip has a high strain constraint; thus, this method is suitable for research on brittle and pseudo-brittle fractures [21]. However, the method cannot be applied to predict ductile fractures when the steel structure does not show obvious defects under a strong earthquake [22]. In 1954, Coffin [23] and Manson [24] introduced an empirical formula, i.e., the Coffin–Manson formula, to predict the LCF life of materials based on the relationship between fatigue life and plastic strain amplitude. However, this method requires several tests to calibrate the necessary parameters and does not consider the effect of triaxiality on fatigue life. In addition, the empirical formula cannot predict the crack path [20,22,25]. A micromechanical model was recently introduced not only to predict the fatigue life of steel structure, but also to describe the effect of stress‒strain field on the microstructural characteristics of the material [26]. Hence, this model can be used to accurately predict the development of ductile cracks in steel structure joints [27]. 

Micromechanical models mainly include the void growth model (VGM) [28], the stress-modified critical strain model (SMCS), the cyclic void growth model (CVGM) [29] and the degraded significant plastic strain model (DSPS) [30]. VGM and SMCS are suitable for calculating the case of monotonic tensile, whilst CVGM and DSPS are used to calculate the case of cyclic loading. In comparison with CVGM, DSPS assumes that the triaxiality of the specimen is constant. Both CVGM and DSPS are semi-empirical and semitheoretical formulas [26] and, thus, require several tests to conduct analysis. The damage degradation parameters of weld specimens are relatively discrete [31]. Moreover, CVGM and DSPS assume that the expansion and contraction rates of microvoids are identical [32], which is inconsistent with actual findings. The location of crack initiation and path cannot be correctly predicted. Finally, CVGM and DSPS cannot directly consider the effect of material damage on material constitutive, and the element size is small, which leads to low calculation efficiency. 

The continuum damage mechanics model (CDM) is another method used to study the fracture properties of steel structures that is applicable to brittle and ductile fractures [33]. By introducing appropriate damage variables, this model can consider the influence of damage on the material constitutive and directly describe the macroscopic mechanical behaviour of the tested materials [34,35]. Moreover, the CDM model can present the law of damage evolution and the post-fracture path [36]. In summary, CDM model has the following advantages: (1) In comparison with the micromechanical model, the CDM model, which is suitable for describing ductile fracture, only needs monotonic tensile test results to calibrate the relevant parameters. This test is simple. In addition, the finite element size is not limited by the characteristic length, and the calculation efficiency is high. (2) The CDM model can predict the ULCF life and fracture position of steel structures as well as the post-fracture path. (3) The CDM model can consider the influence of damage on the material constitutive and can be directly combined with finite element software. Thus, this model can easily be employed in engineering applications and has good development prospects.

In this paper, a refined method to predict the ULCF damage of steel and its welded joints based on the CDM model is studied. The CDM model for monotonic tension is introduced in detail, and the CDM model for ULCF is expanded from this model. Uniaxial tensile tests of Q345qC base metal, heat affected zone (HAZ) and weld metal were carried out to calibrate the material parameters of the CDM model, and a finite element model based on the CDM model was established to predict the fatigue life, fracture life, number of cycles to initial damage and post-fracture path of the specimens. ULCF tests are finally conducted to validate the reliability of the proposed CDM model.

## 2. Theoretical Model for ULCF of Structural Steel

### 2.1. Continuum Damage Mechanics Model for Monotonic Tension

The term damage is used to indicate the deterioration of a material’s capability to carry loads. Damage generally develops in the material microstructure when non-reversible phenomena, such as microcracking, debonding between the matrix and second phase particles and microvoid formation, take place. In the CDM model, the damage degree is generally expressed by the damage variable *D* as follows:(1)$$D=1-\frac{A_{eff}}{A_{0}},$$
where $A_{0}$ and $A_{eff}$ are the nominal and effective cross-sectional area reduced by the presence of microdefects and their mutual interaction, respectively.

In the uniaxial case, the effective stress $\sigma^{{}_{{}^{eff}}}$ is expressed as follows:(2)$$\sigma^{{}_{{}^{eff}}}=\frac{F}{A_{eff}}=\frac{F}{A_{0}(1-D)}=\frac{\sigma }{1-D},$$
where *F* and σ represent the uniaxial tension and tension stress, respectively.

In the framework of the thermodynamics of irreversible phenomena, the constitutive equations of the material can be derived using some state variables [37]. The Helmholtz free energy ψ can be used to characterise a material. The relationship between the Helmholtz free energy and internal variables $x_{i}$ is shown as follows:(3)$$y_{i}=\frac{\partial (\rho \psi )}{\partial x_{i}},$$
where ρ is the material density.

Based on Equations (2) and (3), damage only modifies nominal stress by introducing the concept of effective stress. Hence, elastic effect can be separated from plastic effect.
(4)$$\psi =\psi_{e}(\varepsilon_{ij}^{e},T,D)+\psi_{P}(T,r,\chi ),$$
where $\psi_{e}(\varepsilon_{ij}^{e},T,D)$ and $\psi_{P}(T,r,\chi )$ are the elastic and plastic Helmholtz free energies, respectively, $\varepsilon_{ij}^{e}$ is the elastic strain tensor, T represents the temperature, and r and χ are the hardening variable and kinematic hardening parameter, respectively.

In linear thermoelastic theory, the expression of damage energy release rate is presented as follows [38]:(5)$$Y=\rho \frac{\partial \psi_{e}}{\partial D}.$$

In isotropic materials, the damage energy release rate *Y* can be expressed by Equations (6)–(8):(6)$$Y=-\frac{\sigma_{eq}^{2}}{2E{\left(1-D\right)}^{2}}f\left(\frac{\sigma_{m}}{\sigma_{eq}}\right)$$
(7)$$T=\frac{\sigma_{m}}{\sigma_{eq}}$$
(8)$$f(\sigma_{H}/\sigma_{eq})=2(1+v)/3+3(1-2v){\left(\sigma_{H}/\sigma_{eq}\right)}^{2},$$
where T is the stress triaxiality, $\sigma_{H}$ and $\sigma_{eq}$ present the hydrostatic stress and equivalent von Mises stress respectively, v is Poisson’s ratio and $f(\sigma_{m}/\sigma_{eq})$ is a function of stress triaxiality T.

Based on the Legendre–Fenchel transformation, the dissipative potential $F_{T}$ can be expressed as a function of the related variables [39]$F_{P}\left(\sigma ,R,X;D\right)$ and $F_{D}\left(Y;\varepsilon^{p},D\right)$:(9)$$F_{T}=F_{P}\left(\sigma ,R,X;D\right)+F_{D}\left(Y;\varepsilon^{p},D\right),$$
where *R* and *X* are the isotropic hardening stress and kinematic back stress, respectively, and $\varepsilon^{p}$ is the effective cumulative plastic strain.

The dissipative potential $F_{T}$ in Equation (9) decreases with the classical yield function when the material is not damaged or the damage phenomena associated with void growth are suppressed. For isotropic materials, the first term of Equation (9) can be expressed as in Equation (10). The second term is shown in Equation (11):(10)$$F_{P}\left(\sigma ,R;D\right)=\frac{\sigma_{eq}}{1-D}-R(r)-\sigma_{y}$$
(11)$$F_{D}=\left[\frac{1}{2}{\left(-\frac{Y}{S_{0}}\right)}^{2}\frac{S_{0}}{1-D}\right]\frac{{\left(D_{cr}-D\right)}^{(\alpha -1)/\alpha }}{{\left(\varepsilon^{p}\right)}^{(2+n)/n}},$$
where $\sigma_{y}$ and $S_{0}$ represent the yield stress of the material and the material constant, respectively, α is the damage exponent characteristic of the material, n is the material hardening exponent and $D_{cr}$ represents critical damage variable.

The partial derivation of Equation (11) for *Y* is as follows:(12)$$\frac{\partial F_{D}}{\partial Y}=\frac{Y}{S_{0}}\frac{{\left(D_{cr}-D\right)}^{(\alpha -1)/\alpha }}{(\varepsilon^{p}{)}^{(2+n)/n}}\frac{1}{1-D}$$

Equation (13) can be obtained by substituting Equation (6) into Equation (12):(13)$$\frac{\partial F_{D}}{\partial Y}=-\left(\frac{\sigma_{eq}^{2}}{{\left(1-D\right)}^{2}}\right)f\left(\frac{\sigma_{m}}{\sigma_{eq}}\right)\frac{1}{2ES_{0}}\frac{{\left(D_{cr}-D\right)}^{(\alpha -1)/\alpha }}{{\left(\varepsilon^{p}\right)}^{(2+n)/n}}\frac{1}{1-D}.$$

For ductile materials, the relationship between von Mises equivalent stress $\sigma_{eq}$ and effective cumulative plastic strain $\varepsilon^{p}$ based on the Ramberg–Osgood formula can be obtained as follows [40]:(14)$$\frac{\sigma_{eq}}{\left(1-D\right)}=K{\left(\varepsilon^{p}\right)}^{\left(1/n\right)},$$
where *K* is the material constant.

The kinetic law of damage evolution is expressed by Equation (15). The relationship between the plastic multiplier $\dot{\lambda }$ and the effective accumulated plastic strain rate ${\dot{\varepsilon }}^{p}$ is expressed by Equation (16).
(15)$$\dot{D}=-\dot{\lambda }\frac{\partial F_{D}}{\partial Y}$$
(16)$$\dot{\lambda }={\dot{\varepsilon }}^{p}(1-D)$$

Equations (17) and (18) can be obtained by substituting Equations (14)–(16) into Equation (13):(17)$$\dot{D}=\frac{K^{2}}{2ES_{0}}{\left(D_{cr}-D\right)}^{(\alpha -1)/\alpha }f\left(\frac{\sigma_{m}}{\sigma_{eq}}\right)\frac{{\dot{\varepsilon }}^{p}}{\varepsilon^{p}}$$
(18)$$dD=\frac{K^{2}}{2ES_{0}}{\left(D_{cr}-D\right)}^{(\alpha -1)/\alpha }f\left(\frac{\sigma_{m}}{\sigma_{eq}}\right)\frac{1}{\varepsilon^{p}}d\varepsilon^{p}.$$

Considering proportional monotonic loading, where the stress triaxiality remains constant, Equation (19) can be obtained by integrating Equation (18) between $\left[D_{0},D_{cr}\right]$ and $\left[\varepsilon_{th}^{p},\varepsilon_{f}^{p}\right]$. Similarly, Equation (20) can be obtained by integrating Equation (18) between $\left[D,D_{cr}\right]$ and $\left[\varepsilon_{}^{p},\varepsilon_{f}^{p}\right]$:(19)$${\left(D_{cr}-D_{0}\right)}^{1/\alpha }=\frac{1}{\alpha }\frac{K^{2}}{2ES_{0}}\ln\left(\frac{\varepsilon_{f}^{p}}{\varepsilon_{th}^{p}}\right)f\left(\frac{\sigma_{m}}{\sigma_{eq}}\right)$$
(20)$${\left(D_{cr}-D\right)}^{1/\alpha }=\frac{1}{\alpha }\frac{K^{2}}{2ES_{0}}\ln\left(\frac{\varepsilon_{f}^{p}}{\varepsilon^{p}}\right)f\left(\frac{\sigma_{m}}{\sigma_{eq}}\right),$$
where $\varepsilon_{th}^{p}$ represents the plastic strain threshold under multiaxial stress, $\varepsilon^{p}$ and $\varepsilon_{f}^{p}$ represent the accumulated plastic strain and fracture accumulated plastic strain under multiaxial stress, respectively, and $D_{0}$ is the initial damage variable.

Under uniaxial loading, the stress triaxiality T is equal to 1/3. Equation (8) can then be expressed as follows:(21)$$f\left(\frac{\sigma_{m}}{\sigma_{eq}}\right)=1.$$

The stress triaxiality (*T*) has a weak effect on the plastic strain threshold under multiaxial stress $\varepsilon_{th}^{p}$. Therefore,$\varepsilon_{th}^{p}$ is equal to the plastic strain threshold under uniaxial stress $\varepsilon_{th}^{}$, and $\varepsilon_{f}^{p}$ is equal to the fracture accumulated plastic strain under uniaxial stress $\varepsilon_{f}^{}$. Equation (19) can be expressed as follows:(22)$${\left(D_{cr}-D_{0}\right)}^{1/\alpha }=\frac{1}{\alpha }\frac{K^{2}}{2ES_{0}}\ln\left(\frac{\varepsilon_{f}}{\varepsilon_{th}}\right).$$

Equation (23) can be obtained by substituting Equation (22) into Equation (20).
(23)$$D=D_{0}+(D_{cr}-D_{0})\left\{1-{\left[1-\frac{\ln(\varepsilon /\varepsilon_{th})}{\ln(\varepsilon_{f}/\varepsilon_{th})}\right]}^{\alpha }\right\},$$
where ε represents the accumulated plastic strain under uniaxial stress.

By substituting Equation (22) into Equation (18), Equation (24) can be obtained.
(24)$$dD=\alpha \frac{{\left(D_{cr}-D_{0}\right)}^{1/\alpha }}{\ln\left(\varepsilon_{f}/\varepsilon_{th}\right)}f\left(\frac{\sigma_{m}}{\sigma_{eq}}\right){\left(D_{cr}-D\right)}^{(\alpha -1)/\alpha }\frac{d\varepsilon^{p}}{\varepsilon^{p}}$$

Under proportional loading, Equation (23) can be transformed into Equation (25):(25)$$D=D_{0}+(D_{cr}-D_{0})\left\{1-{\left[1-\frac{\ln(\varepsilon^{p}/\varepsilon_{th}^{p})}{\ln(\varepsilon_{f}/\varepsilon_{th})}f\left(\frac{\sigma_{m}}{\sigma_{eq}}\right)\right]}^{\alpha }\right\}.$$

Equation (26) can be obtained by substituting Equation (22) into Equation (18):(26)$$\frac{\ln(\varepsilon_{f}^{p}/\varepsilon_{th}^{p})}{\ln(\varepsilon_{f}/\varepsilon_{th})}f\left(\frac{\sigma_{m}}{\sigma_{eq}}\right)=1.$$

Considering the weak effect of *T* on the plastic strain threshold, $\varepsilon_{th}^{}$ equals to $\varepsilon_{th}^{p}$. However, the effect of *T* on $\varepsilon_{th}^{p}$ cannot be ignored. Equation (26) can be transformed into Equation (27) as follows:(27)$$\varepsilon_{f}^{p}=\varepsilon_{th}{\left(\frac{\varepsilon_{f}}{\varepsilon_{th}}\right)}^{1/f\left(\frac{\sigma_{m}}{\sigma_{eq}}\right)}.$$

According to the derivation above, Equations (24), (25) and (27) represent the CDM for monotonic tension.

### 2.2. Continuum Damage Mechanics Model for ULCF

The fracture mechanisms for ULCF and monotonic tension involve ductile fracture. Therefore, the CDM for ULCF can be obtained by modifying the CDM for monotonic tension. Under ULCF, when the accumulated plastic tensile strain $\varepsilon^{p+}$ is greater than the uniaxial plastic strain threshold $\varepsilon_{th}^{}$, damage begins to accumulate. Tensile and compressive strain can be distinguished according to the positive or negative sign of stress triaxiality. The CDM calculation (Equation (24)) for monotonic tension can be extended to the field of ULCF, such as in Equations (28) and (29):(28)$$dD=\alpha \frac{{\left(D_{cr}-D_{0}\right)}^{1/\alpha }}{\ln(\varepsilon_{f}/\varepsilon_{th})}f(\frac{\sigma_{m}}{\sigma_{eq}}){\left(D_{cr}-D\right)}^{(\alpha -1)/\alpha }\frac{d\varepsilon^{p+}}{\varepsilon^{p}}$$
(29)$$\{\begin{array}{c} d\varepsilon^{p+}=d\varepsilon^{p}\cdot H(T)\\ H(T)=\{\begin{array}{c} 0\;T<0\\ 1\;T\geq 0 \end{array} \end{array}$$
where $\varepsilon^{p+}$ presents the accumulated plastic tensile strain and H(T) describes the damage state.

Based on the CDM formula for ULCF, the elastic modulus of the material is modified as follows:(30)$$E=E_{0}\left[1-D\cdot H(T)\right]$$
where $E_{0}$ and E represent the elastic modulus before and after damage, respectively.

## 3. Calibration of Material Parameters for CDM

### 3.1. Uniaxial Tensile Test for Notched Round Bar Specimen

Uniaxial tensile tests of Q345qC base metal, heat-affected zone (HAZ) and weld metal are carried out to calibrate the uniaxial plastic strain threshold $\varepsilon_{th}^{}$ in the CDM for ULCF. Here, notched round bar specimens of the base metal, HAZ and weld metal are extracted from the welded steel plate. Different notch radii of specimens result in different stress triaxialities. Three different notch radii (*R* = 4.25, 3.0, 1.5 mm) are prepared for each material in this test, and two specimens are produced for each radius; thus, the total number of specimens is 18. The design dimensions of the specimens are shown in Figure 1, and the number and measurements of the notched round bar specimens are shown in Table 1. A schematic of the loading and measurement procedures for the specimens is shown in Figure 2. The test was carried out at Zhejiang University of Technology. The type of the testing machine is INSTRON-8801 (INSTRON, Norwood, MA, USA).

The load–deformation curve of notched round bar specimens at gauge segment is shown in Figure 3. *P* and *δ* represent the uniaxial tensile load and extension length at the gauge segment, respectively. Under the same radius, the ultimate load-bearing capacity of the weld metal is larger than that of the base metal and HAZ, but the ductility of the former decreases remarkably compared with that of the latter. The ultimate bearing capacity and ductility of the base metal and HAZ are similar.

The abrupt change point of the slope in the descending part of the curve, as shown in Figure 3, reflects the crack initiation point of ductility. During finite element analysis, the extension length at the gauge segment corresponding to the crack initiation point of ductility is taken as the control deformation, which is used to calibrate the uniaxial plastic strain threshold $\varepsilon_{th}^{}$.

### 3.2. Calibration of Material Parameters for CDM by Finite Element Analysis

The two-dimensional axisymmetric finite element model of each notched specimen is established according to the symmetry of the specimen by using ABAQUS (Version 2016, Dassault Systèmes Simulia Corp., Johnston, RI, USA.), as shown in Figure 4. The element type is CAX8R, which is an eight-node quadrilateral biquadratic axisymmetric reduced integral element. The finite element size in the notch area is 0.2 mm, and the central axis of the specimen is subjected to axisymmetric boundary conditions. One end of the notched specimen is articulated, whilst the other end is free in the axial direction.

Uniaxial tension analysis of the finite element model of notched round bar specimens of the three materials is carried out respectively. When the extension length at the gauge segment *δ* in finite element analysis reaches the critical extension length at the crack initiation point of ductility in the test, the stress triaxiality *T* and fracture accumulated plastic strain $\varepsilon_{f}^{p}$ at the centre of the model are recorded. Figure 5 shows the stress–strain field and stress triaxiality nephograms of base metals with different notch radius. The maximum stress triaxiality *T* appears at the centre of the finite element model. At *R* = 4.25 and 3 mm, fracture accumulated plastic strain remains relatively constant throughout the notched section, whilst the maximum fracture accumulated plastic strain appears at the notched surface at *R* = 1.5 mm. Finally, according to *T*, $\varepsilon_{f}^{p}$ and $\varepsilon_{f}$ of the uniaxial tension specimen, the uniaxial plastic strain threshold $\varepsilon_{th}^{}$ can be calibrated by Equation (28).

Table 2 presents the calibration results of $\varepsilon_{th}^{}$. The average uniaxial plastic strain threshold of the base metal and HAZ are 0.4455 and 0.4357, respectively, and show little difference. The $\varepsilon_{th}^{}$ of the weld metal is 0.3673, which is 17.56% and 15.70% less than those of the base metal and HAZ, respectively. The weld metal is the most vulnerable to damage, followed by HAZ and then the base metal. The dispersion coefficients of base metal, HAZ and weld material are 13.75%, 14.91% and 34.84%, respectively, which indicates that the base metal and HAZ materials are more uniform than the weld material. The weld material has the largest dispersion coefficient amongst the specimens tested because of the influence of material uniformity and welding quality.

## 4. ULCF Life Prediction for the Q345qC Base Material and Welded Joints

### 4.1. ULCF Test for Base Material and Welded Joints

#### 4.1.1. Material and Size of Test Specimen

The base material used in the test is Q345qC, which is commonly employed in bridges. A schematic of the extraction of weld metal specimens is shown in Figure 6. The thickness of the steel plate is 32 mm, and the national standard of welding wire is ER50-6. The steel plate is machined into the X-groove perpendicular to the rolling direction, and the welding method is butt welding with a CO_2_ gas shield. The post weld heat treatment method was used to eliminate the influence of welding residual stress on the test result. The sizes of the base and weld metals are shown in Figure 7. The base and weld metal specimens have a circular section with a diameter of 13 mm.

#### 4.1.2. Test Device and Loading System

The test device is an electro-hydraulic servo fatigue testing machine, as shown in Figure 2. Axial strain loading is used in the test. The strain ratio is −1, the loading strain rate is 0.5%/s and the extensometer gauge length is 12.5 mm. Considering the extension measurement range, we set the test loading strain to 7%, 8%, 9% and 10% to obtain ULCF failure of the base and weld metals. Each test was repeated thrice at each loading strain. The numbers of base metal and weld specimens are BMC and WMC, respectively, as shown in Table 3.

#### 4.1.3. Test Results

Figure 8 shows the characteristic curve of the cyclic responses at different strain ranges. Cyclic hardening of the base metal specimen rapidly occurs at the initial stage of the cycle. Subsequently, the degree of cyclic hardening decreases obviously at later stages of the cycle. Finally, the development of cracks causes the stress to decrease sharply until specimen fracture occurs. The welded specimen presents cyclic stability or softening. As shown in Figure 8, the cycle number is defined as the fatigue life of the specimen $N_{f}$ when the stress begins to decrease rapidly. When the specimen fails due to fracture, the cycle number is defined as the fracture life $N_{r}$. The relationship between the fatigue and fracture lives in test is shown in Table 3. The ratio of crack initiation life to fracture life is 84.0–97.5%. Hence, the crack initiation stage occupies most of the cyclic loading period. When the cycle number exceeds the crack initiation life $N_{f}$, the bearing capacity of the specimen decreases sharply and the crack expands rapidly until fracture failure occurs.

The test results for ULCF are shown in Table 4. The average fatigue life of the specimen is *N_f_^’^*. Table 4 shows that the fatigue life of each specimen is fewer than 100 cycles. The fatigue life decreases significantly with increasing strain range. At the same strain range, the fatigue life of weld specimens is only 48–62% that of the base metal specimens. The dispersion of fatigue life of weld specimens is generally larger than that of base metal specimens. The fatigue cracks of base metal specimen are generated from the edge of the section. The fatigue cracks of weld specimen are generated from the edge of HAZ, near the weld area.

The post-fracture of base metal and welded specimens under ULCF is shown in Figure 9. Crack initiation occurs at the edge of the cross section of the base metal specimen in the standard distance section, and the post-fracture path occurs along horizontal direction perpendicular to cross section. The fatigue crack of welded specimens originates from the edge of HAZ near the weld area. Then cracks pass through the weld metal and extend to the HAZ at the opposite side.

### 4.2. ULCF Predictions for Base Material and Welded Joints

#### 4.2.1. Establishment of a Finite Element Model

The two-dimensional axisymmetric finite element model of base metal specimen is established by using ABAQUS, as shown in Figure 10. Figure 11 shows that the weld specimens consist of a base metal, HAZ and weld metal, but welding defects are not considered in the model. Considering both calculation efficiency and accuracy, the multiscale model is adopted for element division. Material and geometric nonlinearities are considered in the 3D finite element model. The material constitutive model is based on the mixed hardening model. Amplitudes, the loading interface provided by ABAQUS, is used to simulate the strain-controlled loading process. Based on Equations (29–30), the subroutine VUMAT of ABAQUS, which is suitable for explicit integration, is used to calculate the damage variable *D*, and the effect of *D* on the elastic modulus *E* is described by Equation (30). *D_0_* and *D_cr_* are equal to 0 and 1, respectively, and α is equal to 0.198 [27]. $\varepsilon_{th}$ is obtained from Table 2.

#### 4.2.2. Fatigue Life Prediction Based on the CDM Model

Figure 12 and Figure 13 show the development of the damage variable *D* for the base metal and weld specimens with numerical simulation. When the cumulative uniaxial plastic strain ε is less than $\varepsilon_{th}$, the base metal and weld specimens are not damaged. Hence, *D* = 0 at the beginning of the cycle. When the cumulative uniaxial plastic strain ε is larger than $\varepsilon_{th}$, the base metal and weld specimen begin to generate damage. Because the damage of both specimens occurs only in the tension stage, and the material parameters are in the exponential position of Equation (29), *D* exponentially increases with a stepped pattern. At the same strain range, the *D* of the cross section centres of the base metal and weld specimens reach *D_cr_* earlier than that of the corresponding cross section edges. The crack predicted by CDM originates from the cross section centre for the base metal and weld specimens. In addition, CDM can predict the number of cycles to initial damage *N_0_*. At the same strain range, the number of cycles to initial damage *N_0_* at the cross section centres for the base metal and weld specimens is less than those at the corresponding cross section edges. As the total strain range increases, the fatigue life *N_f_* and numbers of cycles to initial damage *N_0_* of the base metal and welded specimens decrease gradually. According to Equation (29), the growth rate of the damage variable dD is proportional to the cumulative plastic strain increment $d\varepsilon^{p+}$. Hence, a large total strain range leads to a large dD and $d\varepsilon^{p+}$, which causes low *N_0_* and *N_f_*.

Comparison of the predicted crack initiation and fracture lives of the base metal and weld specimens with test results are shown in Table 5 and Table 6. $N_{f,C}^{P}$ and $N_{f,E}^{P}$ represent the predicted fatigue life at the cross section centre and edge, respectively. $N_{f}$ and $N_{r}$ represent the test fatigue and fracture life, respectively. As shown in Figure 12 and Figure 13, cracks appeared from the centre to the edge. Therefore $N_{f,C}^{P}$ and $N_{f,E}^{P}$ can be considered the predicted crack initiation life and predicted fracture, life respectively. Table 5 shows that the predicted values of the crack initiation and fracture lives of base metal specimens are less than the corresponding experimental values. The predicted errors of crack initiation life and fracture life decrease with increasing strain level. Indeed, the larger the strain range in the test, the more consistent the ULCF damage of the base metal specimens with ductile failure characteristics.

Table 6 compares the predicted crack initiation and fracture lives of weld specimens with test results. The table shows that the predicted fatigue and fracture lives of weld specimens are less than the corresponding test values. Errors in the predicted crack initiation and fracture lives of welded specimens decrease with increasing strain level.

The post-fracture path of base metal specimens at strain range of 7% under ULCF load are shown in Figure 14. As the cycle number increase, elements satisfying the fracture criterion (*D* = *Dcr*) are deleted individually. When the crack width reaches half the size of the specimen, the difference between cycle number and fracture life is very low at only two or three times. Hence, when the initial cracks generate, the cracks develop very fast. Cracks develop from the cross section centre to the edge, and their direction of development is perpendicular to the axis direction of the specimen. The fracture surface is relatively flat. In addition, the predicted post-fracture path is identical to the test result, as shown in Figure 9a.

The post-fracture path of weld specimens at a strain range of 7% are shown in Figure 15. Cracks in weld specimens appear at the cross section centre. With an increasing cycle number, cracks propagate from the cross section centre to the edge of the weld zone. Then, the cracks reach the narrow area of the weld zone on the specimen surface and then develop along the interface between the weld metal zone and HAZ. Finally, oblique cracks are formed, and the weld specimen completely fractures. The post-fracture path of weld specimens is demonstrated by the test results in Figure 9b.

According to the above analysis, CDM can successfully predict fatigue life, fracture life and post-fracture path, as well as the number of cycles to initial damage. The predicted crack initiation location and post-fracture path are similar to the results for weld metal specimens, but the predicted crack initiation location for the base metal specimen deviates from the test results to some extent. The possible reasons are as follows: (1) In the CDM model of ULCF, the material is assumed to have no defects; however, a few inclusions or defects in the specimens are generated in production stage; and (2) a CDM that is suitable for ULCF only considers the damage caused by tensile strain, thus neglecting effect of compressive strain on damage.

## 5. Conclusions

The CDM model is introduced in this paper to predict the ULCF damage of steel and its weld joints in serious earthquakes. Firstly, a CDM that is suitable for ULCF damage, extending from the case of monotonic tension, is proposed. The material parameters of the CDM model are then calibrated using uniaxial tensile tests. Moreover, finite element models based on the CDM model are established using ABAQUS to predict the fatigue life, fracture life, number of cycles to initial damage and post-fracture path of the specimens. The tests of ULCF for base metal and weld specimens to validate the accuracy of the CDM model. According to the above research, the following conclusions can be drawn:

(1) Considering that the fracture mechanisms of ULCF and monotonic tension are ductile fracture, the CDM model for ULCF damage is reasonably extended from the CDM model for monotonic tension. The CDM model for ULCF damage only considers the damage of tensile strain and ignores the damage of compressive strain.

(2) The uniaxial plastic strain threshold $\varepsilon_{th}^{}$ of Q345qC base metal, HAZ and weld metal is calibrated. Amongst the samples, the weld metal has the lowest $\varepsilon_{th}^{}$, and HAZ has a slightly lower $\varepsilon_{th}^{}$ than the base metal. The weld metal is the most vulnerable to damage, followed by HAZ and the base metal.

(3) The poor uniformity of the weld metal and the quality of welding exert important effects on $\varepsilon_{th}^{}$. In comparison with the $\varepsilon_{th}^{}$ of the base metal and HAZ, the weld metal specimen had the largest dispersion coefficient about $\varepsilon_{th}^{}$.

(4) The development law of the damage variable *D* reveals exponential growth combined with a stepped pattern.

(5) The predicted crack initiation and fracture lives of the base metal specimens are lower than the corresponding experimental values. The predicted errors of crack initiation life and fracture life decrease with an increasing strain level. The CDM model can accurately predict the number of cycles to initial damage. In addition, as the strain range increases, the fatigue life, fracture life and number of cycles to initial damage decrease.

(6) The predicted crack initiation location in the weld specimen is the centre of the cross section, whilst the location of the test is the HAZ edge. The predicted crack initiation location in the base metal specimen is the centre of the cross section, whilst the location of the test is the edge. The predicted crack initiation location slightly differs from the location of the test. Two reasons may explain these results. Firstly, the CDM model assumes that no defects exist in the steel; however, a few inclusions or defects are generated in production stage. Secondly, the CDM of ULCF only considers the damage caused by tensile strain and neglects the effect of compressive strain.

(7) The direction of post-fracture path in the base metal specimen is perpendicular to the axis direction. The post-fracture path in the welded specimen occurs along the interface between the weld metal zone and HAZ, and oblique cracks are finally formed. The predicted post-fracture path is identical to the test results.

## Figures and Tables

**Figure 1 materials-12-04014-f001:**
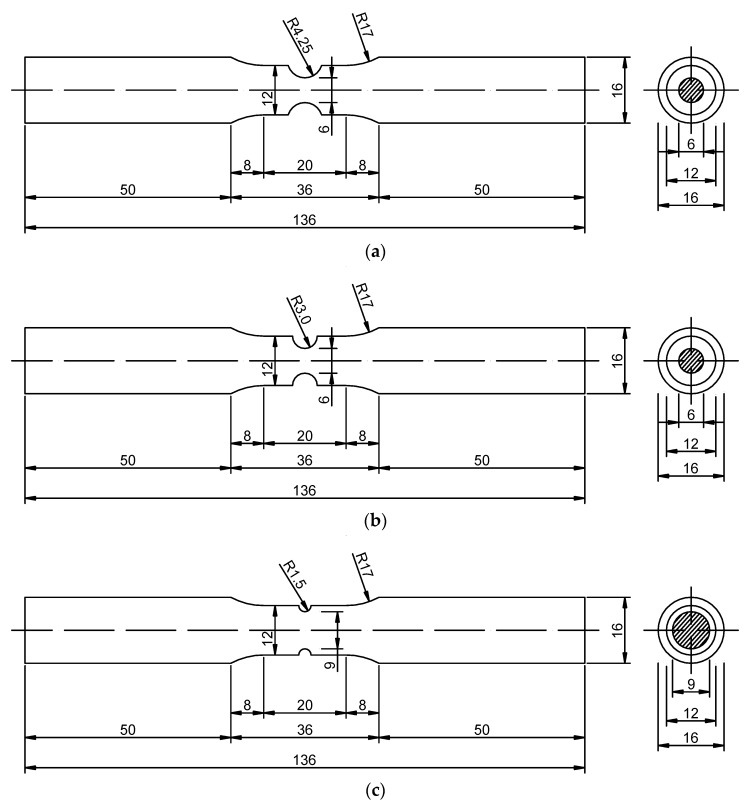
Designed size of notched round bar specimens with notch radii of (**a**) *R* = 4.25 mm, (**b**) *R* = 3.0 mm, and (**c**) *R* = 1.5 mm.

**Figure 2 materials-12-04014-f002:**
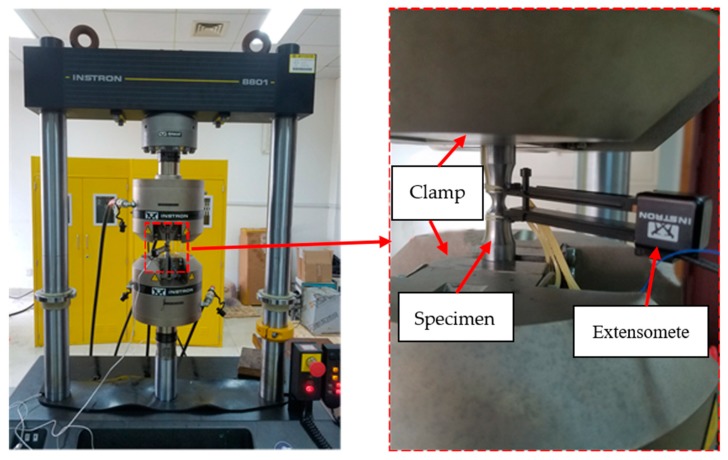
Schematic of the loading and measurement procedures for notched round bar specimens.

**Figure 3 materials-12-04014-f003:**
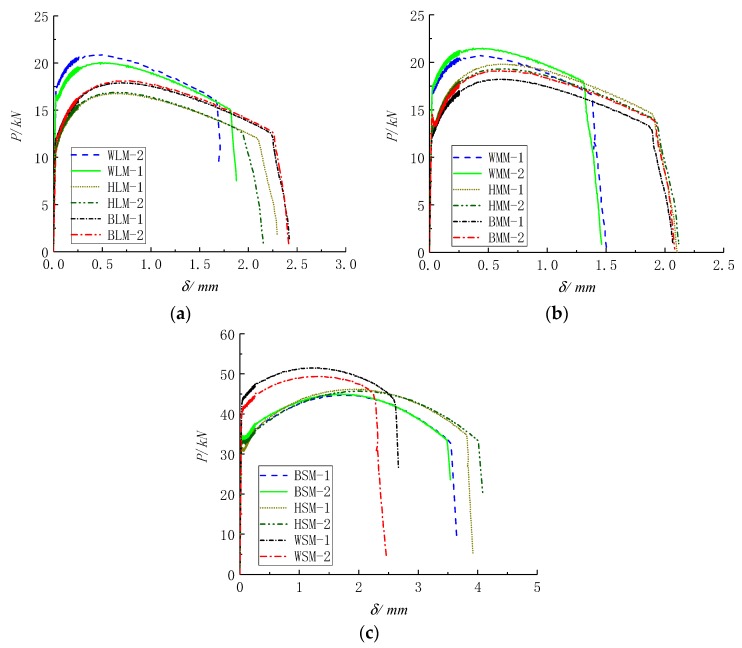
Load–deformation curves of notched round bar specimens at the gauge segment with notch radii of (**a**) R = 4.25 mm, (**b**) R = 3.0 mm, and (**c**) R = 1.5 mm.

**Figure 4 materials-12-04014-f004:**
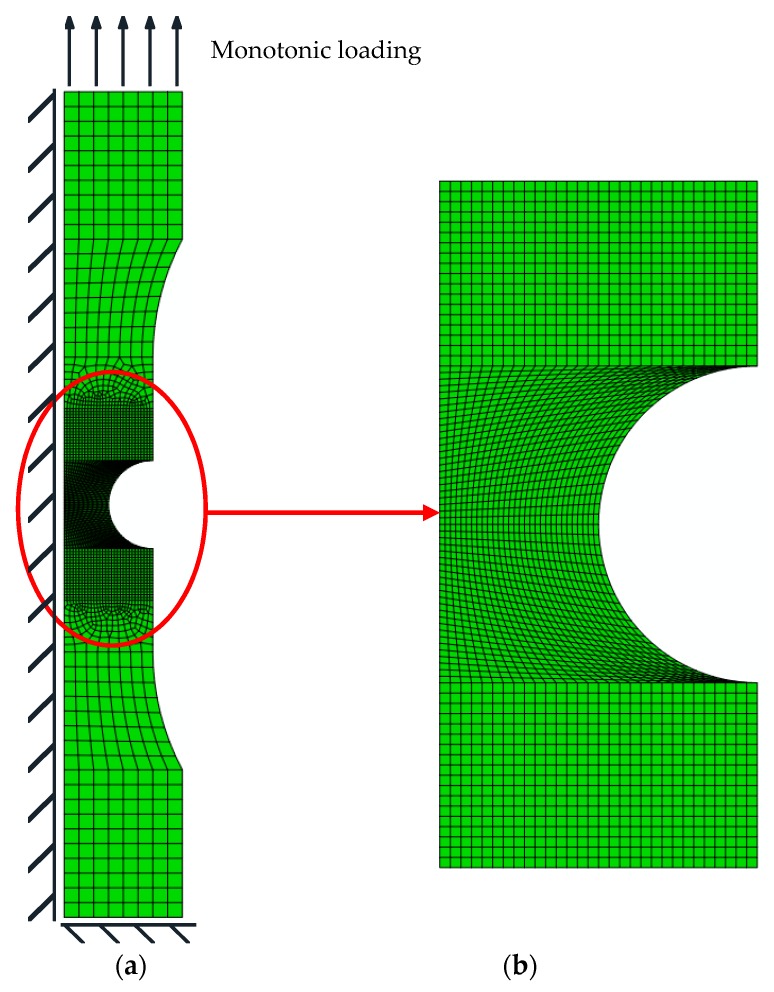
Two-dimensional axisymmetric finite element model of a notched round bar specimen. (**a**) Integral finite element diagram of a notched round bar specimen. (**b**) Local finite element diagram of a notched round bar specimen.

**Figure 5 materials-12-04014-f005:**
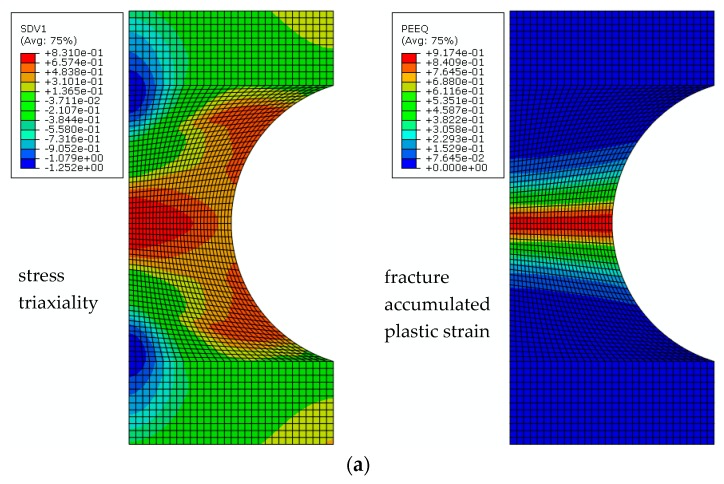
Stress triaxiality and fracture-accumulated plastic strain nephograms of the base metal with notch radii of (**a**) *R* = 4.25 mm and (**b**) *R* = 1.5 mm.

**Figure 6 materials-12-04014-f006:**
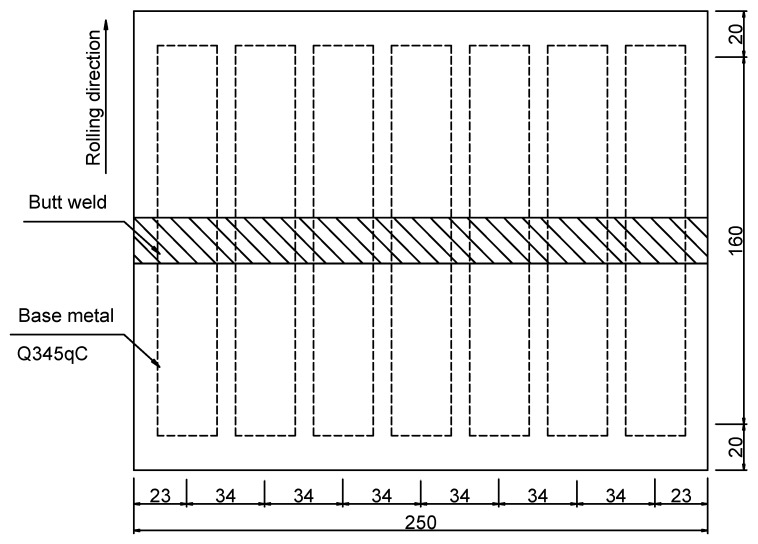
Schematic of the extraction of the weld metal specimen (unit: mm).

**Figure 7 materials-12-04014-f007:**
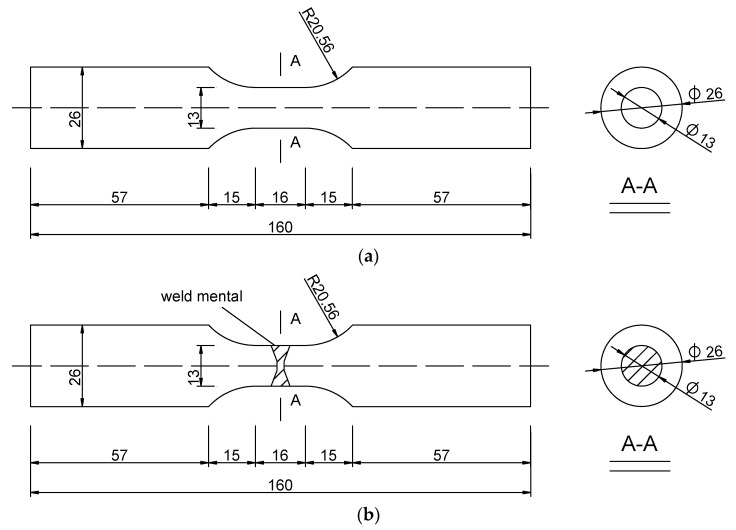
Size of specimens for ULCF (unit: mm). (**a**) Base metal specimen and (**b**) weld specimen.

**Figure 8 materials-12-04014-f008:**
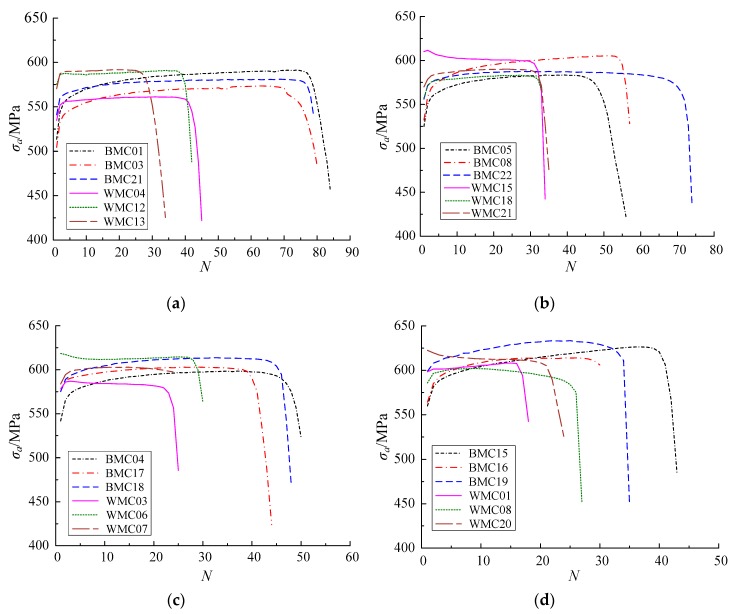
Characteristic curve of cyclic responses at strain ranges of (**a**) 7.0%, (**b**) 8.0%, (**c**) 9.0% and (**d**) 10.0%.

**Figure 9 materials-12-04014-f009:**
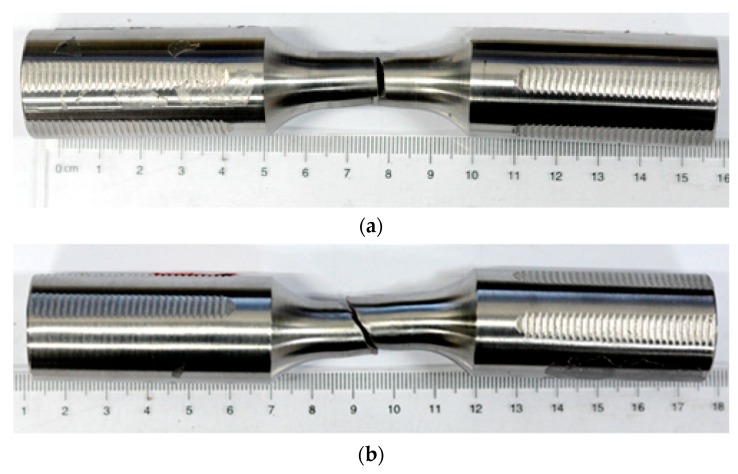
Post-fracture path of ULCF: (**a**) Base metal specimen and (**b**) weld specimen.

**Figure 10 materials-12-04014-f010:**
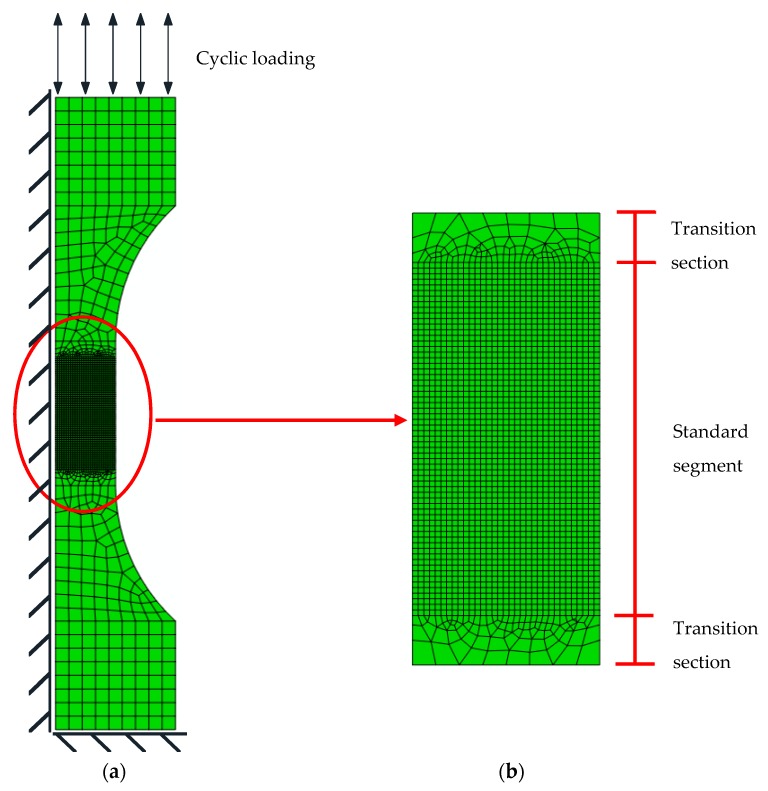
Axisymmetric finite element model of the base metal specimen: (**a**) Integral finite element model and (**b**) local finite element model.

**Figure 11 materials-12-04014-f011:**
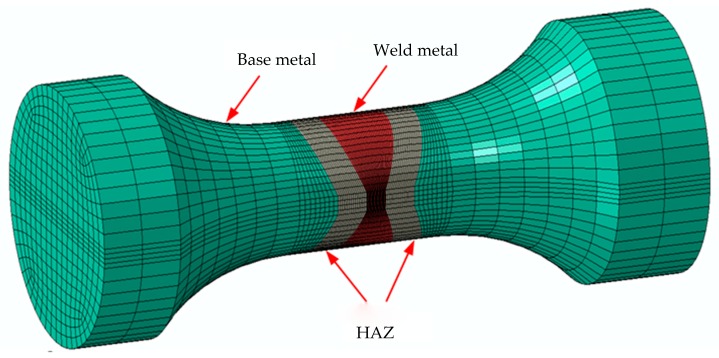
3D finite element model of the weld specimen.

**Figure 12 materials-12-04014-f012:**
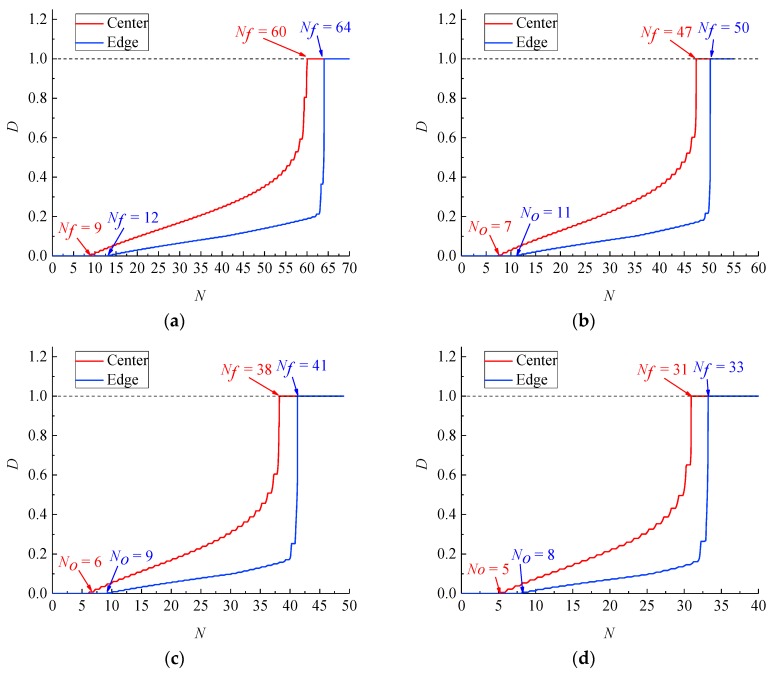
Development of the damage variable *D* of the base metal specimen at strain ranges of (**a**) 7%, (**b**) 8%, (**c**) 9% and (**d**) 10%.

**Figure 13 materials-12-04014-f013:**
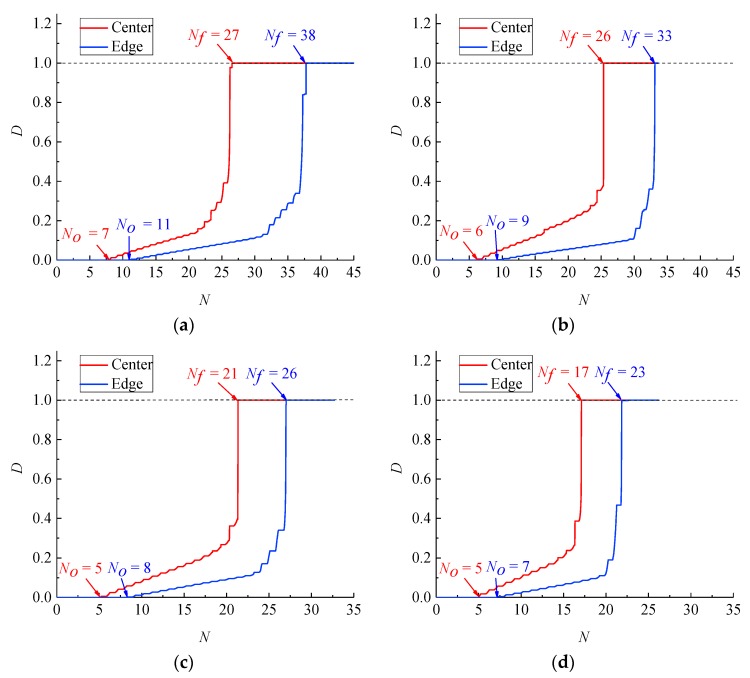
Development of the damage variable *D* of the weld specimen at strain ranges of (**a**) 7%, (**b**) 8%, (**c**) 9% and (**d**) 10%.

**Figure 14 materials-12-04014-f014:**
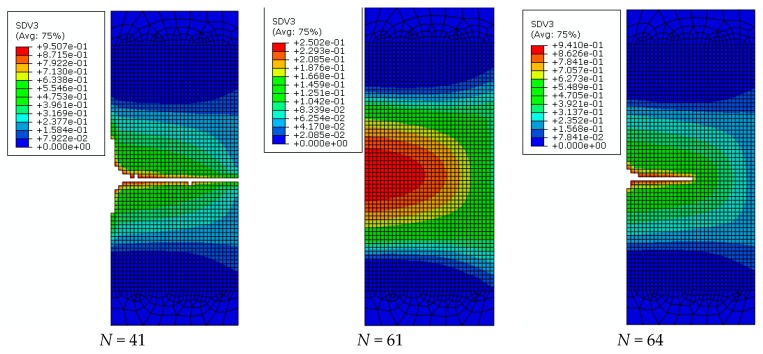
Development of cracks in the base metal specimen (strain range: 7%).

**Figure 15 materials-12-04014-f015:**
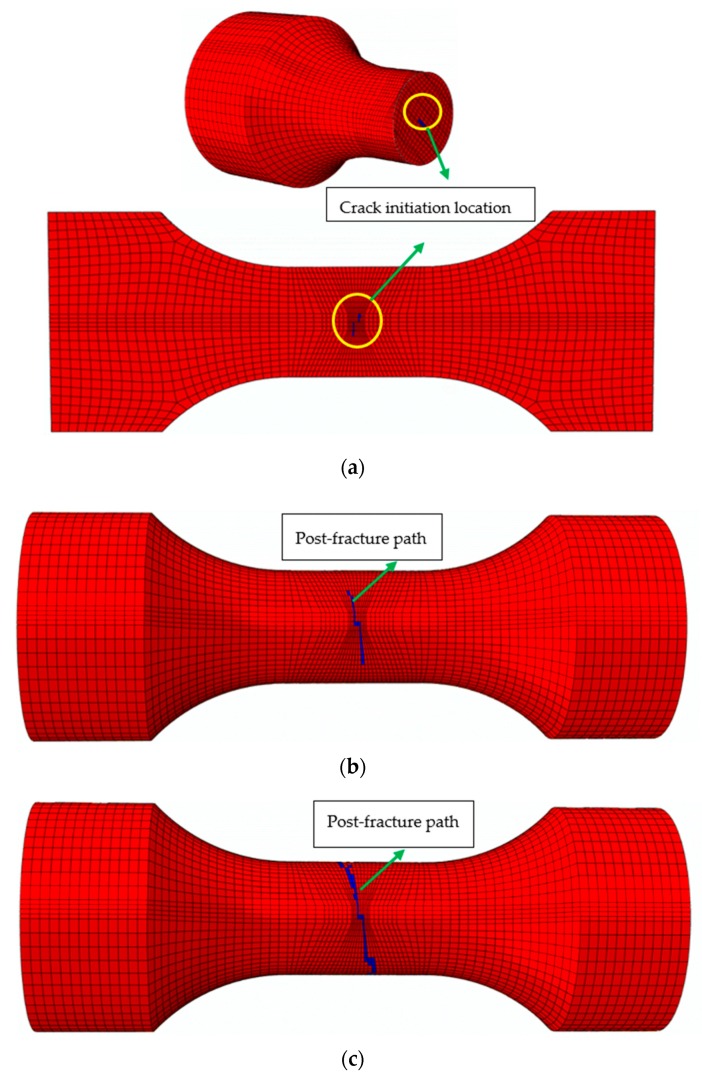
Development of cracks in the weld specimen (strain range, 7%). (**a**) *N* = 27, (**b**) *N* = 35 and (**c**) *N* = 38.

**Table 1 materials-12-04014-t001:** Number and measured size of notched round bar specimens.

Material	Notch Radius (mm)	Number	Clamping Segment Diameter (mm)	Gauge Segment Diameter (mm)	Intermediate Segment Length (mm)	Notch Root Diameter (mm)
Base metal	4.25	BLM-1	15.95	11.96	35.98	6.04
		BLM-2	15.95	11.92	36.09	6.04
	3.0	BMM-1	15.99	11.94	35.73	5.97
		BMM-2	16.02	11.95	35.79	6.04
	1.5	BSM-1	15.94	11.9	36.02	9.08
		BSM-2	15.97	11.89	35.90	9.04
Heat affected zone	4.25	HLM-1	15.99	12.01	36.16	5.85
		HLM-2	15.97	11.97	36.18	5.79
	3.0	HMM-1	15.99	12.00	35.94	6.40
		HMM-2	15.95	11.93	35.89	6.24
	1.5	HSM-1	15.99	12.01	36.37	9.35
		HSM-2	16.04	11.96	35.99	9.35
Weld metal	4.25	WLM-1	15.96	12.02	36.02	6.33
		WLM-2	16.00	11.91	36.53	6.25
	3.0	WMM-1	15.97	11.92	36.03	6.24
		WMM-2	15.94	11.91	36.37	6.29
	1.5	WSM-1	15.99	12.04	36.40	9.43
		WSM-2	16.05	11.96	36.27	9.36

**Table 2 materials-12-04014-t002:** Calibration results of material parameters for CDM.

Material	*R* (mm)	Number	*δ_f_* (mm)	$\varepsilon_{f}$	T	$\varepsilon_{f}^{p}$	$\varepsilon_{th}$
Base metal	4.25	BLM-1	2.240	1.3186	0.8310	0.8893	0.5047
		BLM-2	2.262	1.3186	0.8337	0.9119	0.5387
	3.0	BMM-1	1.876	1.3186	0.8916	0.7882	0.4210
		BMM-2	1.917	1.3186	0.8920	0.8153	0.4541
	1.5	BSM-1	3.543	1.3186	1.089	0.6634	0.3895
		BSM-2	3.477	1.3186	1.096	0.6366	0.3648
	Average						0.4455
	Standard deviation						0.0613
	Dispersion coefficient (Standard deviation / Average)						13.75%
Heat affected zone	4.25	HLM-1	2.098	1.3084	0.8205	0.8262	0.4179
		HLM-2	1.934	1.3084	0.8145	0.7380	0.3111
	3.0	HMM-1	1.895	1.3084	0.8862	0.7963	0.4310
		HMM-2	1.922	1.3084	0.8865	0.8087	0.4464
	1.5	HSM-1	3.808	1.3084	1.0572	0.7760	0.5035
		HSM-2	4.009	1.3084	1.0570	0.7768	0.5044
	Average						0.4357
	Standard deviation						0.0649
	Dispersion coefficient (Standard deviation / Average)						14.91%
Weld metal	4.25	WLM-1	1.821	0.9882	0.8629	0.7813	0.5736
		WLM-2	1.672	0.9882	0.8595	0.7109	0.4591
	3.0	WMM-1	1.371	0.9882	0.9430	0.6221	0.3790
		WMM-2	1.306	0.9882	0.9470	0.5904	0.3419
	1.5	WSM-1	2.593	0.9882	1.2070	0.4540	0.2805
		WSM-2	2.264	0.9882	1.2082	0.3329	0.1700
	Average						0.3673
	Standard deviation						0.1280
	Dispersion coefficient (Standard deviation / Average)						34.84%

**Table 3 materials-12-04014-t003:** Relationship between fatigue and fracture lives in the test.

Δ*ε_t_*/%	Number	*N_f_*	*N_r_*	*N_f_*/*N_r_*
7	BMC01	78	84	92.9%
	BMC03	76	81	93.8%
	BMC21	78	80	97.5%
	WMC04	43	46	93.5%
	WMC12	40	42	95.2%
	WMC13	29	34	85.3%
8	BMC05	51	57	89.5%
	BMC08	55	57	96.5%
	BMC22	72	75	96.0%
	WMC15	32	35	91.4%
	WMC18	32	33	97.0%
	WMC21	33	36	91.7%
9	BMC04	47	51	92.2%
	BMC17	41	45	91.1%
	BMC18	45	48	93.8%
	WMC03	24	26	92.3%
	WMC06	29	31	93.5%
	WMC07	24	25	96.0%
10	BMC15	41	44	93.2%
	BMC16	30	31	96.8%
	BMC19	33	35	94.3%
	WMC01	17	19	89.5%
	WMC08	25	27	92.6%
	WMC20	21	25	84.0%

**Table 4 materials-12-04014-t004:** Test results for ULCF.

Δ*ε_t_*/%	Number	Crack initial location	Fatigue life *N_f_*	Average *N_f_^’^*	Standard deviation	Dispersion coefficient
7.0	BMC01	Section edge	78	77	0.94	0.01
	BMC03	Section edge	76			
	BMC21	Section edge	78			
	WMC04	HAZ edge	43	37	6.02	0.16
	WMC12	HAZ edge	40			
	WMC13	HAZ edge	29			
8.0	BMC05	Section edge	51	59	9.10	0.15
	BMC08	Section edge	55			
	BMC22	Section edge	72			
	WMC15	HAZ edge	32	32	0.47	0.01
	WMC18	HAZ edge	32			
	WMC21	HAZ edge	33			
9.0	BMC04	Section edge	47	44	2.49	0.06
	BMC17	Section edge	41			
	BMC18	Section edge	45			
	WMC03	HAZ edge	24	25	2.36	0.09
	WMC06	HAZ edge	29			
	WMC07	HAZ edge	24			
10.0	BMC15	Section edge	41	34	4.64	0.13
	BMC16	Section edge	30			
	BMC19	Section edge	33			
	WMC01	Weld edge	17	21	3.27	0.16
	WMC08	Weld edge	25			
	WMC20	HAZ edge	21			

**Table 5 materials-12-04014-t005:** Comparison of the predicted crack initiation and fracture lives of base metal specimens with test results.

*Δεt* %	$N_{f,C}^{P}\;(\mathrm{Predicted})$	*Nf* (Test)	Relative Error (%)	$N_{f,E}^{P}\;(\mathrm{Predicted})$	*Nr* (Test)	Relative Error (%)
7.0	60	77	−22.1	64	82	−21.9
8.0	47	59	−20.3	50	63	−20.6
9.0	38	44	−13.6	41	48	−14.6
10.0	31	35	−11.4	33	37	−10.8

**Table 6 materials-12-04014-t006:** Comparison of the predicted crack initiation and fracture lives of weld specimens with test results.

*Δεt* %	$N_{f,C}^{P}\;(\mathrm{Predicted})$	*Nf* (Test)	Relative error (%)	$N_{f,E}^{P}\;(\mathrm{Predicted})$	*Nr* (Test)	Relative error (%)
7.0	27	37	−27.0	38	41	−7.3
8.0	26	32	−18.8	33	35	−5.7
9.0	21	26	−19.2	26	27	−3.7
10.0	17	21	−19.0	23	24	−4.2

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
