# Peer review of "Prediction of the Ultra-Low-Cycle Fatigue Damage of Q345qC Steel and its Weld Joint"

_materials, 2019, doi:10.3390/ma12234014_

Round 1
Reviewer 1 Report
The manuscript under the title: Prediction of the ultra-low cycle fatigue damage of Q345qC steel and its weld joint” is partly coherent with the Materials journal. The authors join computer modelling with the experimental research work. The organization of the article is appropriate. Overall, the paper is well prepared. Nevertheless, the article required some improvements:
the introduction part MUST BE up-dated. The proposed literature is quite old. There is lack of citation new articles. Only 4 cited articles have been published after 2016 (less than 10%), the article required extensive editing. The figures: 3, 4, 5, 7, 8, 11, 12, 14, 15 are partly outside the editing area. please add the comments under the table 2. please verify the interline in text. please give the type of the testing machine.Author Response
Please see the attachment.

Reviewer 2 Report
Prediction of the ultra-low cycle fatigue damage of 2 Q345qC steel and its weld joint
Many thanks to the authors for this interesting manuscript. It deals with the prediction of the Ultra-Low Cycle Fatigue in steel piers under seismic actions. My comments are listed below:
1) I would include more details in the abstract regarding the results and the innovative findings.
2) line 33 42 what is the novelty in this paper? Is the ULCF or the prediction of the failure in steel piers?
3) The literature reviewed in the introduction is broad and cover several aspects. However it does not cover the criteria for the definition of the fatigue failure. Also, since you are studing welded structures it should be discussed in the introduction.
4) line 99 theoretical model for ULCF, it is not clear whether the model is yours or you are taking from other sources. References might be needed.
5) The theoretical model discussed up to line 212 is similar to the Manson-Coffin approach. Is the triaxiality the difference?
6) Figure 1 should have the axes to show that the specimens are cylindrical
7) for some reasons some of the figures and the text goes beyond the limit of the pages and it is not clear.
8) is the calibration discussed from line 297 for the static or the fatigue loading?
9) the comparison of the strain to failure should be detailed in terms of locations where it is measured.
10) The diagram in figure 6 shows a plate but the samples are shown in the next figure with a cylindrical cross section. The technical drawing should be amended and corrected to follow the conventional standard. Axes are required when the shape is cylindrical.
11) The comparison between experimental and numerical results should also include pictures of the tested specimens to show that the numerical model matches the experimental results. Also, the graphical results like those in figure 8 should be used to compare the FEA.
12) I would suggest to structure the manuscript separating the experimental and the numerical results as it is now confusing. Does Figure 12 refer to experimental or numerical results?
13) There are different FE models used throughout the manuscript and it is not clear the reason for the different approaches. Is figure 14 a cross section of the 3D model?
14) It would be useful to discuss the mesh sensitivity, the location of the failure and the crack path in more details and more clearly.
15) any effects due to the residual stresses? How did you account for it?
16) It would help to use the D-N curves shown in Figure 13 to report the comparison between experimental and numerical model.
As you are considering the triaxial effects in your model, the crack path and the prediction is crucial. The manuscript should have more details.
